# Recognition of BRAF by CDC37 and Re-Evaluation of the Activation Mechanism for the Class 2 BRAF-L597R Mutant

**DOI:** 10.3390/biom12070905

**Published:** 2022-06-28

**Authors:** Dennis M. Bjorklund, R. Marc L. Morgan, Jasmeen Oberoi, Katie L. I. M. Day, Panagiota A. Galliou, Chrisostomos Prodromou

**Affiliations:** 1Biochemistry and Biomedicine, School of Life Sciences, University of Sussex, Falmer, Brighton BN1 9QG, UK; dennis.bj90@gmail.com; 2Department of Life Sciences, Faculty of Natural Sciences, South Kensington Campus, Imperial College London, London SW7 2AZ, UK; rhodri.morgan@imperial.ac.uk; 3Genome Damage and Stability Centre, School of Life Sciences, University of Sussex, Falmer, Brighton BN1 9RQ, UK; j.oberoi@sussex.ac.uk; 4Domainex, Pampisford, Cambridge CB2 3EG, UK; katie.day@domainex.co.uk; 5Laboratory of Biological Chemistry, Aristotle University of Thessaloniki, 54124 Thessaloniki, Greece; ag.gal.work@gmail.com

**Keywords:** Hsp90, CDC37, BRAF, kinase, activation mechanism, chaperone, co-chaperone

## Abstract

The kinome specific co-chaperone, CDC37 (cell division cycle 37), is responsible for delivering BRAF (B-Rapidly Accelerated Fibrosarcoma) to the Hsp90 (heat shock protein 90) complex, where it is then translocated to the RAS (protooncogene product p21) complex at the plasma membrane for RAS mediated dimerization and subsequent activation. We identify a bipartite interaction between CDC37 and BRAF and delimitate the essential structural elements of CDC37 involved in BRAF recognition. We find an extended and conserved CDC37 motif, ^20^HPNID---SL--W^31^, responsible for recognizing the C-lobe of BRAF kinase domain, while the c-terminal domain of CDC37 is responsible for the second of the bipartite interaction with BRAF. We show that dimerization of BRAF, independent of nucleotide binding, can act as a potent signal that prevents CDC37 recognition and discuss the implications of mutations in BRAF and the consequences on signaling in a clinical setting, particularly for class 2 BRAF mutations.

## 1. Introduction

Specific protein kinases are known to exist as an ensemble of conformations due to their metastable state [1,2] and it is this instability that likely defines their dependency on the Hsp90-CDC37 (heat shock protein 90 – cell division cycle 37) complex. CDC37 is a kinase specific co-chaperone that delivers protein kinases, such as BRAF (B-Rapidly Accelerated Fibrosarcoma), to the Hsp90 complex [3]. CDC37 consists of three domains, an n-terminal domain, linked by a beta-strand to the middle domain and finally a small c-terminal helical domain. Several incomplete structures of CDC37 have, to date, been determined [1,4,5,6,7]. However, molecular details on how CDC37 recognizes client kinases is still poorly understood. The cryo-EM structure of the Hsp90-CDC37-Cdk4 (Cdk4, cyclin-dependent kinase 4) complex (PDB 5FWK and EMD-3337), shows that the c-terminal lobe of the Cdk4 kinase domain is engaged with the n-terminal domain of CDC37 and in particular with the base of the c-terminal helix of CDC37 that makes up the helix coiled-coil structure. Specifically, a small loop (HPNI) from CDC37 mimics a similar loop (HPNV in Cdk4 and HVNI in BRAF) found in the n-terminal lobe of kinases [5], which is normally engaged in binding to a helix in the c-terminal lobe of kinase domains. The middle domain of CDC37, as seen in the cryo-EM structure, is also potentially involved in interactions with the n-terminal lobe of kinase domains [5]. Finally, a helix linker leads to the c-terminal domain of CDC37, the function of which is less clear and not visible in the cryo-EM structure of the Hsp90-CDC37-Cdk4 complex [5].

Protein kinases play a central role in regulating eukaryotic signaling pathways in key processes, such as cell survival, metabolism, proliferation, cell migration and differentiation and in the cell cycle [8]. As such, their regulation is of paramount importance, and dysregulation of their activity can lead to cell transformation and cancer [9]. Kinases have been described as molecular switches that can adopt at least two extreme conformations, in which the catalytic machinery through conformational remodeling becomes correctly aligned for catalysis to take place. The maximally active conformation is known as the “on” state, while the inactive state is referred to as the “off” state [9]. In particular, the DFG loop of kinases is conformationally flexible and adopts either an inactive (out) or an active (in) conformation [10], which helps to remodel the rest of the kinase for activity. For example, in the active ‘DFG-in’ position of epidermal growth factor receptor (EGFR; PDB 5CNO), the phenylalanine residue forms part of the R--(regulatory) spine, which consists of a series of hydrophobic residues that connect essential elements required for catalysis to the F-helix, while the ATP, once bound, helps to complete the C-(catalytic) spine, a series of residues that connect the F-helix to the n-terminal lobe of the kinase domain [11,12,13]. In the inactive (‘out’) state of EGFR (PDB 2RF9) the same phenylalanine of the DFG motif is displaced from the R-spine and positions itself so that the C-spine is disrupted [11]. Thus, this phenylalanine is positioned in such a way as to clash with the phosphate groups of the bound ATP. Clearly, the conformational regulation of kinases from their inactive (‘out’) state to their ‘in’ or active state is of paramount importance for attaining their enzymatic activity. BRAF mutants that drive oncogenesis consist of three classes [14]. Class 1 BRAF mutants, which consist of Val 600 mutations, signal as RAS (protooncogene product p21)-independent active monomers, where dimerization is disrupted, and are insensitive to ERK1/2 and SOS feedback inhibition. Class 2 mutants function as dimers, but their activation appears to be RAS-independent. Hence, they escape feedback inhibition through the phosphorylation of SOS, a modification that downregulates its activity. Finally, class 3 mutations are kinase impaired but increase signaling through the MAPK (Mitogen-Activated Protein Kinase) pathway, due to enhanced RAS binding and subsequent CRAF activation.

The Hsp90-CDC37 complex is required for the stability of BRAF and, as such, both active (V600E mutant) and inactive forms of the kinase act as clients [15,16,17]. For wild type BRAF, binding to the Hsp90-CDC37 complex ultimately leads to its delivery at the plasma membrane for interaction with RAS, where BRAF is subsequently activated. This RAS-BRAF complex is more abundant than the Hsp90-CDC37 complex of BRAF [18]. The classical mechanism for BRAF activation occurs by its association with RAS and 14-3-3 at the cellular membrane, together with Hsp90 and CDC37. Activation of BRAF leads to translocation of the cytoplasmic Hsp90-CDC37-Braf complex to the cell membrane [18], while inhibition of Hsp90 by geldanamycin leads to a rapid dissociation of both Hsp90-BRAF and RAS-BRAF multimolecular complexes, increased proteasomal degradation of BRAF and a decrease in translocation of BRAF to the plasma membrane [17,19,20,21]. In the absence of RAS the most populated state appears to be the autoinhibited form of BRAF that is further stabilized by interaction with 14-3-3, which binds directly to the phosphorylated amino acid residues Ser (pSer) 365 and 729 of BRAF [22]. In the presence of activated RAS, however, there is a shift from the inactive state to an active BRAF signaling complex [23]. The general consensus is that the interaction of RAS with the inactive BRAF-14-3-3 complex displaces 14-3-3 from pSer 365 allowing either homodimerization or heterodimerization of BRAF [24], which remains tethered through the pair of pSer 729 residues of the BRAF dimer to a dimer of 14-3-3. In addition, by RAS promoting dimerization of BRAF this allows the cis- autophosphorylation of Thr 599 and Ser 602 of the BRAF activation loop [24,25]. Such phosphorylation is critical in inducing and stabilizing a conformational change that leads to alignment of the C- and R-hydrophobic spines of the kinase domain, thus promoting ATP uptake and, consequently, MEK (Mitogen-Activated Protein Kinase (MAPK) kinase) phosphorylation [11,26]. In contrast to wild type BRAF, it was recently observed that the Hsp90-CDC37-BRAF V600E complex was not only more abundant than the 14-3-3 complex, but was shown to be more active [16]. Thus, there appears to be an altered partitioning between the 14-3-3-RAS and the Hsp90-CDC37 complex caused by the V600E mutation in BRAF.

The current work explored the interaction between CDC37 and a variety of BRAF mutants, and defined the CDC37 and BRAF domains involved. We aimed to understand how BRAF mutations cause partitioning between the cytosolic Hsp90-CDC37 complex and the membrane bound 14-3-3—RAS complex. We present evidence that dimerization of BRAF, as seen with the class 2 L597R BRAF mutation [27], severs recognition by the kinome specific CDC37-dependant co-chaperone of Hsp90. We propose that the Hsp90-CDC37 chaperone system may play a regulatory role in maintaining the class 1 BRAF V600E mutant, while the class 2 mutant BRAF L597R assembles into a dimeric structure that is resistant to interaction with CDC37. Consequently, a disequilibrium results to the overall regulatory pathways that tightly govern the normal activity of BRAF and we discuss the consequences of such dysregulation.

## 2. Material and Methods

### 2.1. Protein Expression and Purification

Constructs of human c-terminally His-tagged CDC37 cloned in pET28b and n-terminally GST-tagged sBRAF kinase domain (residues 423–723) cloned in p3E (A. W. Oliver, University of Sussex), including mutant forms, were obtained from Genscript for expression in Escherichia coli BL21 (DE3) pLysS by induction at 20 °C with 1 mM isopropyl-1-thio-β-D-galactopyranoside MERCK, catalogue No. I6758, Darmstadt, Germany. CDC37 was expressed as previously described using Talon metal affinity chromatography (Takara Bio company, catalogue No. 635652, Saint-Germain-en-Laye, France), Superdex 75 or 200 PG gel-filtration and Q-sepharose ion-exchange [28]. sBRAF kinase domain constructs were purified using glutathione affinity resin (Genscript, catalogue. No. L00206, Rijswijk, Netherlands), followed by PreScission cleavage overnight, and Superdex 200 gel-filtration. All concentration steps utilized Vivaspin 30 centrifugal concentrators (Sartorius, catalogue number VS2022, Goettingen, Germany). Purified proteins were dialyzed against 20 mm Tris/HCl (MERCK, Calbiochem, catalogue No. 648317, Darmstadt, Germany), pH 7.5, containing 1 mm EDTA (MERCK, Sigma Aldrich, catalogue No. E5134-1KG, Darmstadt, Germany), and 200 mM NaCl (MERCK, Sigma Aldrich,, catalogue No. S9888-5KG, Darmstadt, Germany), in preparation for isothermal titration chromatography (ITC).

### 2.2. Isothermal Titration Chromatography Kd Determinations

Heat of interaction was measured on an ITC_200_ microcalorimeter (Malvern) under the same buffer conditions (20 mM Tris, pH 7.5, containing 1 mM EDTA and 200 mM NaCl). In most cases, aliquots of sBRAF construct at 350 μM were injected into 30 μM of CDC37 at 20 °C. Heats of dilution were determined by diluting injectant into buffer. Data were fitted using a curve-fitting algorithm (OriginLab Cooperation, Microcal Origin, version 7.0, Northhampton, MA, USA).

### 2.3. BRAF Kinase Assays

The activity of sBRAF was determined by a MEK phosphorylation assay consisting of 35 μM sBRAF, 6mM MgCl_2_ (MERCK, CAlbiochem, catalogue No. 442611-M, Darmstadt, Germany), 5 mM ATP (MERCK, Sigma Aldrich, catalogue No. A7699-5G, Darmstadt, Germany), 1 μg inactive MEK1 (c-terminally His-tagged) in a total volume of 40 μL buffer (20 mM Hepes pH 8 (MERCK, Sigma Aldrich, catalogue No. 54457-250G-F, Darmstadt, Germany), 1 mM DTT (MERCK, Sigma Aldrich, catalogue No. D9779-25G, Darmstadt, Germany, 100 mM NaCl). Reactions were incubated at 30 °C for 60–180 min and samples taken for western blot analysis. Phophorylated MEK was detected using anti-phospho MEK 1/2 (residues 218/222 and 222/226, MERCK, catalogue No. 05-747, Darmstadt, Germany) antibody and rabbit secondary HRP (Cyvita, catalogue No. NA934-1ML, Marborough, MA, USA) both at a 1/5000 dilution. Detection was carried out using a Pierce ECL Western Blotting Substrate (Thermo Fisher Scientific, catalogue No. 32106, Waltham, MA, USA).

### 2.4. Thermal Shift Assay

Reactions were carried out in triplicate using an Applied Bioscience, StepOnePlus real time PCR system. Experiments were conducted using 2 μM of sBRAF or mutant protein in a total of 20 μL of 20 mM Tris pH 7.5, 1 mM EDTA, 1 mM DTT and 200 mM NaCl containing 2.5 μL of 1/250 diluted SYPRO orange (Applied Biosystems Protein Thermal Shift, Thermo Fisher Scientific, Catalogue No. 2023-08-31, Waltham, MA, USA). The temperature was ramped up from 14 to 90 °C for over 60 min with an integration time of 1 sec. Data was analyzed with the LightCycler 480 Software version 1.5. (Roche, Basel, Switzerland).

### 2.5. Molecular Mass Determination

Samples of 50 μL of protein were loaded onto a Superdex 200 Increase 10/300 GL gel-filtration column equilibrated in 50 mM HEPES pH 7.5, 300 mM NaCl, 0.5 mM TCEP (MERCK, catalogue No. C4706, Darmstadt, Germany and 1 mM EDTA. The gel-filtration standards used were β-amylase (200 kD), aldolase (158 kD), conalbumin (75 kD), ovalbumin (44 kD) and carbonic anhydrase (29 kD) from a combination of two kits (GE Healthcare, catalogue No. 28-4038-41, Chicago, Illinois, IL, USA and MERCK, Sigma Aldrich, catalogue No. MWGF1000, Darmstadt, Germany).

## 3. Results

### 3.1. The Binding of CDC37 to sBRAF and sBRAF V600E Is Essentially Indistinguishable

Many kinases are notoriously difficult to produce in *E. coli*. In this study, we used a solubilized version of BRAF kinase domain (sBRAF, residues 423-723). This has a series of surface mutations that help to solubilize the kinase and improve yields in *E. coli* [29]. Previously, work showed that the co-expression of Hsp90, CDC37 and sBRAF in insect cells resulted in a stable Hsp90-CDC37-kinase complex. Such complexes containing sBRAF or sBRAF V600E were indistinguishable from that formed with native (unsolubilized) BRAF kinase domain [17]. Both wild type and sBRAF kinase domains were also shown to bind CDC37 *in vitro*, forming a stable complex in gel-filtration.

We found that both sBRAF and sBRAF V600E bound CDC37 with similar affinities (sBRAF, *K*d = 1.0 μM and sBraf V600E *K*d = 0.41 μM, Figure 1A,B), confirming earlier studies [17] that the solubilizing mutations on the surface of the kinase, as well as the V600E mutation, did not significantly disrupt the CDC37-BRAF interaction. Since BRAF V600E is more prevalent in Hsp90-CDC37 complexes [16] we chose the sBRAF V600E mutant, which gives reasonable yields following purification from *E. coli*, to further study CDC37 binding. Binding of sBraf V600E was promoted by an enthalpic change resulting from the interaction (−12,670 cal/mol), which was offset by an unfavorable entropy (−14 cal/mol/°C), indicating some degree of order resulting from the interaction (Figure 1B). This would be consistent with CDC37 stabilizing a dynamically unstable kinase domain.

### 3.2. Nucleotide Binding Prevents the CDC37-BRAF Interaction

We next tested the effect of the bound nucleotide, which binds deep in a pocket formed by the N- and c-terminal lobes of the kinase domain, on the binding of CDC37 to sBRAF V600E. We found that both AMPPNP and ADP prevented association of CDC37 with the kinase (Figure 1C,D), which was consistent with earlier observations [17]. These results support the idea that the CDC37 interaction with BRAF is focused within the nucleotide binding pocket of the kinase, as observed in the cryo-EM structure of the Hsp90-CDC37-Cdk4 complex [5].

### 3.3. The n-terminal and c-terminal Domains of CDC37 Are Essential for Efficient Bipartite Binding to the Kinase Domain

The interaction between CDC37 and kinases remains enigmatic and we wanted to better understand how kinases are recognized by CDC37. Based on structural studies [4,5] (Figure 2A), three CDC37 constructs, representing the n-terminal-(residues 1–120), the middle-(residues 148 to 269) and the c-terminal-domains (residues 273–353), were expressed and ITC interaction studies conducted to determine which domains of CDC37 were required for the sBRAF V600E interaction. We found that the n-terminal (amino acid residues 1–120) and c-terminal domain (residues 273 to 348) of CDC37 were both compromised in their ability to bind sBRAF V600E (*K*d = 278 and 104 μM, respectively; Figure 2B,C) relative to full-length CDC37 (*K*d = 0.41 μM; Figure 1B). In contrast, the middle domain failed to interact all together (Figure 2D). This contrasts with the Cryo-EM structure of Hsp90-CDC37-Cdk4, where the c-terminal domain of CDC37 was not visible [5].

### 3.4. Determining the Minimal CDC37 Structure Required for High-Affinity Binding to sBRAF V600E

In order to delimitate the exact segments that could be deleted from CDC37, but still maintained high affinity binding, we made a series of mutants that lacked structural elements of the CDC37 structure. Figure 2A shows structural elements and sites for CDC37 modification in ITC studies used throughout this study and Appendix A shows a graphical summary of all the constructs made. We first removed a large section of the coiled-coil (CC) structure (residues 44 to 108) from the n-terminal domain of full-length CDC37 and replaced it with a shorter tryptophan zipper (TrpZip) sequence (GSWTWENGKWTWKSG; CDC37-TrpZip) [30]. This sequence allowed the formation of a simple β-hairpin with stable secondary structure and could, therefore, preserve the structure of the remaining n-terminal domain. CDC37-TrpZip was soluble and bound sBRAF V600E with normal affinity (*K*d = 0.5 μM, Figure 3A). This suggested that the coiled-coil structure of the n-terminal domain was superfluous for kinase binding by CDC37 alone. We next shortened the remaining coiled-coil region by introduction of a small linker (Gly-Ser-Gly) between residues 41 and 111, and within the same construct deleted the β-strand (βS) immediately following the n-terminal domain, by introducing a Gly-Ser-Gly linker between residues 119 and 140 to create CDC37-Δ(CC-βS). Using ITC, we tested this construct for binding to sBRAF V600E and found that it essentially bound normally (*K*d = 0.61 μM, Figure 3B). This suggested that, in addition to the CC region, (residues 42–110), the βS element (residues 120 to 139) that links the n-terminal and middle domains of CDC37 was indispensable for kinase binding. Next, we deleted within the CDC37-Δ(CC-βS) construct the first 7 n-terminal residues and residues 272 to 285, which represent a small piece of helix that joins the middle and c-terminal-domain and terminated the protein at position 348 (to create CDC37-Δ(7-CC-βS-[272-285])). CDC37-Δ(7-CC-βS-[272-285]) bound sBRAF V600E normally (*K*d = 1.03 μM; Figure 3C), relative to intact CDC37 (*K*d = 0.41 μM; Figure 1B). Next, we created a construct (CDC37-Δ(7-CC-βS-[120-285]-[349-378]); abbreviated N^m^C) that essentially linked the N and C-domains of CDC37 together, but maintained the deletions from the previous construct. CDC37-N^m^C, resulted in a substantial reduction in affinity for sBRAF V600E (*K*d = 145 μM; Figure 3D) to a level previously seen for the individual N- and C-domains of CDC37 (Figure 2B,C). We, therefore, reasoned that the spatial distance between the N- and C-domains was probably the reason for the reduced affinity. Consequently, we reintroduced 14 amino acid residues back into CDC37-N^m^C to create CDC37-N^m^(+14)-C. Residues 272 to 285 represented a piece of the helix that links the middle- and c-terminal domains of CDC37 and had previously been shown not to be required for direct binding to sBRAF V600E. CDC37-N^m^(+14)-C was found to have substantially increased binding affinity to sBRAF V600E (*K*d = 8.7 μM; Figure 4A) relative to N^m^C (*K*d = 145 μM; Figure 3D). Similarly, by reintroducing residues 120-125, representing a piece of the βS that links the n-terminal and middle-domains, into CDC37-N^m^C to create CDC37-N^m^(+6)-C, some binding affinity was restored (*K*d = 18 μM; Figure 4B). It would, thus, appear that the binding affinity of CDC37 and sBRAF V600E was influenced by the spatial separation of the N- and c-terminal domains. We tested this further by then reintroducing residues 245 to 285, which represent a long helix that links the middle and C-domain of CDC37, into CDC37-N^m^C to create CDC37-N^m^(+41)-C. This construct showed vastly improved binding affinity for sBRAF V600E (*K*d = 1.99 μM; Figure 4C) relative to CDC37-N^m^C (*K*d = 145 μM; Figure 3D). The results suggested that there is a minimal spatial distance between the CDC37 N- and c-terminal domains that is required for efficient kinase binding.

We next investigated the limits required for binding by the c-terminal domain. The structure of the c-terminal domain of human CDC37 has been previously determined by both X-ray crystallography and by solution NMR [4,6] (PDB, 1US7 and 2N5X, respectively). These structures reveal a small domain that appears to be structurally mobile and Trp 342 is an essential residue of the core required for the folding of the domain. In contrast, valine at position 343 appears not to be fully packed in the core of either structure and its hydrophobic side-chain remains exposed to solvent. CDC37, consisting of residues 1 to 343, bound sBRAF V600E normally (*K*d = 0.76 μM; intact CDC37; *K*d = 0.41 μM) (Figure 1B and Figure 5A, respectively). Similarly, the V343A and V343R mutants were more or less normal for sBRAF V600E binding (*K*d = 0.9 and 1.3 μM, respectively; Figure 5B,C). However, CDC37 1-342 was compromised in its affinity for sBRAF V600E (*K*d = 45.7 μM; Figure 5D). This suggests, that Val 343 may aid Trp 342 packing and is essential for folding of the c-terminal domain, but is not, in itself, essential for direct binding of kinases. Consistent with this was that W342A binding to sBRAF V600E was compromised (*K*d = 94.3 μM, Figure 5E), whereas, and as expected, the W342R construct failed to bind altogether (Figure 5F), suggesting that Trp 342 is essential for the folding of the c-terminal domain of CDC37.

The results so far suggested that the n-terminal domain (residues 8 to 41, which contains the HPNI motif), residues 111 to 119 (which immediately follow the CC region) and residues 286 to 348 (which represent the c-terminal domain) are the minimal elements required for high affinity binding to kinases, although a non-specific spacer (we used residues 245 to 285) between the N-domain and C-domain elements is also required (Figure 5G). Collectively, the results suggested that the interaction between CDC37 and kinases is bipartite and that the lobes of a kinase domain are probably held in a spatially separated conformation.

### 3.5. The CDC37 S13E-Phosphomimetic Mutation Does Not Affect Binding to BRAF

It has been reported that phosphorylation of Ser 13 within CDC37 is required for the binding of a kinase and that the S13E mutation can act as a phosphomimetic [31]. However, the cryo-EM structure of Hsp90-CDC37-Cdk4, suggests that Ser 13 is not directly involved in the interaction with the kinase [5]. To test this hypothesis, we compared the binding of sBRAF V600E to CDC37 S13E and CDC37 wild type. We found that the affinity for the binding of the S13E mutant to be similar to unmutated CDC37, (*K*d = 0.30 and 0.41 μM, respectively; Figure 1B and Figure 6A). This was consistent with the cryo-EM structure of Hsp90-CDC37-Cdk4 complex that showed that phosphoserine is involved in a serious of contacts involving Lys 406 of Hsp90 and His 33 and Arg 36 of CDC37, but does not contact Cdk4 directly (Figure 6B).

### 3.6. The CDC37 HPNI Amino Acid Motif Is Essential for High-Affinity Kinase Recognition

Analysis of the Hsp90-CDC37-Cdk4 cryo-EM structure suggests that the c-terminal lobe of Cdk4 is recognized by a conserved amino acid motif, HPNI [5], which mimics the HPNV motif interactions within Cdk4 and HVNI within BRAF. In kinases this motif, present in the N-lobe of these kinases, forms part of the normal packing interactions between the two lobes of the kinase domain. Closer inspection showed that the side-chain of Asn 22, of the CDC37 HPNI motif, may be involved in polar contacts with the side-chains of Thr 153 and Arg 126 and a main-chain carbonyl interaction with Val 154 of Cdk4 (Figure 7A). To test the importance of the HPNI motif we mutated Asn 22 to alanine and arginine. As expected, both the CDC37 N22A and N22R mutants were compromised for binding to sBRAF V600E (*K*d = 27.3 and *K*d = 14.2 μM, respectively; Figure 7B,C). This confirmed that the HPNI motif of CDC37 is involved in the initial recognition of kinases.

### 3.7. HPNI Is Part of a More Extensive CDC37 Binding Motif

On closer inspection of the CDC37 structure we noticed that the recognition of the kinase was potentially more extensive and involved a conserved motif consisting of ^20^HPNID---SL--W--Q^34^), of which the I----SL--W sequence formed mostly a small hydrophobic patch or pocket that could form interactions with a bound kinase (Figure 8A). In our analysis, amino acid position 23 was generally a conserved Ile or Val, position 24 was a conserved Asp, position 27 was a conserved Ser, position 28 was either a Leu or Phe and position 31 was a conserved Trp. We therefore decided to test whether the mutation of these conserved residues of CDC37 would influence kinase binding. As expected, we found that the L28R mutation completely abolished the interaction with sBRAF V600E (Figure 8B), while the L28A mutant diminished it substantially (*K*d = 16.9 μM; Figure 8C), relative to wild type CDC37 (*K*d = 0.41 μM; Figure 1B). This was consistent with the L28R mutation causing a steric clash that prevented sBRAF V600E from binding. In contrast, the L28A mutation did not prevent binding, but compromised the strength of the interaction seen. Similarly, the interaction of W31K with CDC37 was abolished. (Figure 8D), but the W31A mutation was well tolerated (*K*d = 2.1 μM; Figure 8E). Furthermore, the mutation S27K reduced binding substantially (*K*d = 29 μM; Figure 8F), whereas the S27A mutation was well tolerated *K*d = 1.6 μM; Figure 8G). In contrast to these mutants, the Q34R and Q34A mutations had very little effect on sBRAF V600E binding (*K*d = 0.3 and 0.35 μM, respectively; Figure 8H,I). However, we noted that arginine was found at position Gln 34, in some CDC37 proteins. These results, collectively, suggested that the motif for the recognition of the c-terminal lobe of kinase domains could be extended to include ^20^HPNID---SL--W^31^, although experimentally we did not test Asp 24, since the side chain of this amino acid residue is critical for maintaining the structure of the HPNI loop by forming polar interactions with the side-chain of Ser 27 and the main-chain amide of Ala 26 at the base of the proceeding helix.

### 3.8. Activation Loop Mutants Do Not Influence BRAF Binding to CDC37

BRAF is activated through phosphorylation of two key residues, T599 and S602 within the activation loop or T loop of kinases. In addition, the V600E mutation is known to constitutively activate BRAF, and is the most prominent mutation in cancer [32]. We have previously seen that the sBRAF V600E mutation does not affect binding to CDC37, but we wondered if the phosphomimetic mutations T599E and S602D would affect Cdc37 binding. The sBRAF T599E-V600E-S602D mutant was previously shown to be catalytically active [17]. Using this active triple mutant in ITC experiments we showed that binding to CDC37 was unaffected (*K*d = 0.47 μM; Figure 9A). Furthermore, mutating Thr 599 for a bulky tryptophan, or a positively charged arginine side-chain, did not affect binding to CDC37 (*K*d = 0.23 μM; Figure 9B). The results showed that phosphomimetic mutations of the T loop did not influence CDC37 binding.

### 3.9. The BRAF Mutation L597R Compromises Binding to CDC37

The conformation of the DFG motif of kinases is known to affect kinase activity and numerous mutations have been documented that alter these residues and are associated with oncogenic phenotypes. We, therefore, decided to analyze such oncogenic mutations and their effect on CDC37 binding. Unlike V600E, the F595V mutation in BRAF is inactivating [33] and consequently, to prevent complicated scenarios, we decided to make all activation segment mutants in a V600 background. We found that both the sBRAF F595A and sBRAF F595V mutations did not compromise binding to CDC37 (*K*d = 0.5 and 1.4 μM, respectively; Figure 10A,B). Similarly, for sBRAF D594V, where D594V results in impaired kinase activity [34,35,36], we found this construct bound CDC37 normally (*K*d = 1.2 μM; Figure 10C).

We next tested the binding of sBRAF L597A and sBRAF L597R mutants in a Val 600 background. We found that the L597A mutation did not compromise binding to CDC37 (*K*d = 0.23 μM; Figure 10D), whereas we were surprised to find that the L597R mutation abolished binding completely (Figure 10E). In EGFR the equivalent mutation, L858R (L834R in mature EGFR), has been shown to form salt bridges with Glu 758 (BRAF Ala 497) and Glu 762 (BRAF Glu 501) or Glu 761 (BRAF Asn 500), which are found on the dynamic regulatory element, known as the C-helix [26,37]. Another polar residue on the C-helix of BRAF is also found at Gln 496. A structural analysis of the BRAF structure (PDB 4RZV) suggests that L597R may allow salt bridges, among other possible interactions, with the nearly invariant Glu 501, which is located on the C-helix. Glutamate at this position within kinases forms a salt bridge with the invariant Lysine (BRAF Lys 483), which itself becomes coupled to bound ATP. However, a salt bridge between L597R and Glu 501 (or other possible residues) would stabilize the C-helix of sBRAF, which is directly connected, at its c-terminal end, to the αC-β4 loop containing the HPNI residues that CDC37 mimics in binding to kinases. This loop is highly significant as it is the only element from the n-terminal lobe of kinase domain that is functionally and constitutively connected to its c-terminal lobe.

It has been previously shown for the equivalent mutation in EGFR, L858R destabilizes the inactive kinase, but simultaneously stabilizes the C-helix and that this consequently leads to dimerization of the mutant kinase [38,39,40,41]. In order to test the stability of the L597R mutation, we conducted a thermal shift assay. We compared the thermal stability of L597R with sBRAF and various other sBRAF mutants (Figure 11A). As expected, we found that inactive sBRAF was the most stabile construct with a Tm = 40 °C and that sBraf V600E was significantly less stabile (Tm = 36.2 °C). However, the L597A mutation destabilized the kinase further (Tm = 35 °C). In contrast, the L597R mutation (Tm = 37.8 °C) was less stable than sBRAF, but more stable than sBRAF V600E. This was in agreement with the destabilizing effect of this mutation on the inactive kinase, whilst simultaneously stabilizing the C-helix, which consequently leads to dimerization of the kinase. The L597R-V600E double mutant Tm (36.8 °C) was found to be in between the V600E Tm (36.2 °C) and the Tm for sBRAF L597R (36.8 °C).

Due to the tendency for EGFR L858R to promote dimerization, we next investigated the oligomeric nature of the equivalent BRAF L597R mutation. It was found that sBRAF had a relative molecular mass of 35.5 kD, as eluted from a Superdex 200 Increase 10/300 GL column, which was similar to its calculated relative molecular mass of 33.4 kD (Figure 11B,C). In contrast, L597R displayed a relative molecular mass of 60.2 kD, which was consistent with a dimeric state for the kinase domain. BRAF V600E eluted with a relative molecular mass of 34.7 kD similar to that of sBRAF, while the double mutant, L597R V600E, reverted the relative molecular mass of L597R back towards the monomeric state (38.9 kD). Collectively, the results suggested that L597R drives BRAF into a dimeric state that is not recognized by CDC37, and that V600E drives the kinase back to a monomeric state.

We next asked whether we could disrupt the potential L597R-Glu 501 salt bridge, as the most likely, but not the only interaction stabilizing the dimerization promoting C-helix conformation. Introducing the E501A mutation into a L597R background we found that the L597R-Glu 501 mutant expressed poorly and was unstable. However, we had previously seen that V600E could disrupt dimerization of L597R and, therefore, further work was conducted with this double mutant, L597R-V600E. We found that introducing the V600E mutation into a L597R background completely restored binding with CDC37 (*K*d = 0.34 μM; Figure 11D), suggesting that the V600E mutation had destabilized the L597R dimeric promoting component of the double mutant.

### 3.10. sBRAF L597R and L597R-V600E Mutant Are Both Inactive

Literature suggests that L597R is an activating mutation [42,43]. However, on closer inspection the results suggested this is inferred indirectly by determining cellular levels of phospho-MEK and phospho-ERK [44,45]. Our *in vitro* MEK phosphorylation assays showed that L597R was inactive relative to V600E (Figure 11E). Thus, it appeared that dimerization alone of L597R was insufficient, under the conditions used, for the activation of its kinase activity. We next assayed the BRAF L597R-V600E mutant for kinase activity. We found that, although V600E restored CDC37 binding, V600E did not activate kinase activity in the double mutant (Figure 11F), which suggested that further mechanisms were required to attain an activated L597R mutant conformation or signaling complex.

## 4. Discussion

CDC37 is a kinome-specific co-chaperone responsible for delivering protein kinases to Hsp90. The molecular details of how CDC37 recognizes such kinases has remained enigmatic, but recent work suggests that kinases are structurally dynamic and, consequently, require stabilization by CDC37 [1,2]. We have now shown that CDC37 recognizes kinases by a bipartite interaction involving two CDC37 elements, which was recently suggested by NMR and cryo-EM studies [5]. We defined a small n-terminal CDC37 fragment, ^20^HPNID---SL--W^31^, and its c-terminal domain, as the interaction sites for kinases. The interacting fragments must be connected by a minimal distance, which we achieved by introducing residues 245–285 of the native CDC37 sequence, which normally forms an α-helix, but appeared to be otherwise dispensable for binding. The interaction of the c-terminal domain of CDC37 with the kinase was not observed in the high-resolution cryo-EM structure of Hsp90-CDC37-Cdk4 [5]. This suggested that CDC37-kinase complex may be remodeled after interaction with Hsp90. A bipartite interaction between CDC37 and BRAF was recently reported using NMR [46], but here we more clearly defined residues involved in that interaction. The authors of the NMR study suggested that CDC37 needs to be in a compact form to recognize kinase protein. We found that by replacing the central regions of CDC37 with smaller linkers, which connect the N- and c-terminal domains of CDC37, we could still observe binding between CDC37 and sBRAF. The longest linker, of 41 amino acids, appeared to, more or less, restore full binding, (*K*d = 1.99 μM; Figure 4C). However, shorter linkers of 14 and 6 amino acids also displayed significant binding (*K*d = 18 and 8.7 μM; Figure 4A,B, respectively). This suggested some flexibility in the exact conformation of CDC37 for its bipartite recognition of the kinase, which might reflect an ensemble of kinase conformations, due to their dynamic instability.

Mutational analyses suggest that the n-terminal domain of CDC37 uses an extended motif (^20^HPNID---SL--W^31)^, which incorporates the conserved sequence HPNI, to recognize the c-terminal lobe of kinases. In contrast, mutation of the activation segment residues did not affect CDC37 binding, except for the L597R mutation. The equivalent L858R mutation, in EGFR tyrosine kinase, drives a variety of cancers including non-small-cell lung cancer [47,48,49,50,51]. As with EGFR L858R [37,38,39,40], we found L597R also drives dimerization of BRAF.

The Hsp90-CDC37 complex has been reported to be present at the plasma membrane with RAS and, presumably, can deliver BRAF to the RAS complex [18]. The conformation of the kinase, altered by mutation, is likely to influence this process. We suggest that the active class 1 BRAF V600E mutant, although recognized by CDC37, is less able to enter the RAS complex, as entry into this complex normally occurs as a BRAF autoinhibited complex. This could explain why elevated levels of cytoplasmic Hsp90-CDC37-BRAF V600E accumulate above those normally seen for wild type BRAF (Figure 12). In the case for the class 2 BRAF L597R mutant, CDC37 is unable to complex with the already formed dimer of this mutant, which means that elevated levels of BRAF L597R may accumulate in the cytoplasm and enhanced signaling may be established, following activation of such a dimer (Figure 12). In our hands, dimerization of the BRAF L597R mutant was not sufficient to bring about its activation. We also found that the V600E mutation was unable to activate the BRAF L597R mutant. Thus, the activation of BRAF L597R might require other mechanisms that bring about a catalytically active state for this mutant, in whatever form it is signaling.

We propose that the BRAF V600E mutant preferentially partitions to the Hsp90-CDC37 complex and that Hsp90 most likely maintains the cytoplasmic stability of this active mutant by preserving the V600E kinase against proteasomal degradation. It has, of course, been well documented that the V600E mutant is sensitive to Hsp90 inhibition [52]. However, what are the implications of our findings for BRAF L597R signaling? For the L597R mutant, Hsp90-CDC37 may promote dimerization of the kinase by stabilizing initially the monomeric form, soon after translation, which ultimately leads to its dimerization, following the normal cycles of Hsp90 chaperone activity. However, it remains unclear how such a dimer becomes active, as inclusion of V600E in the L597R background failed to activate the double mutant. This raises a number of questions. Firstly, does activation of the L597R mutant require full phosphorylation of the activation loop, where V600E alone is insufficient to bring about activity? Secondly, if phosphorylation alone does not activate L597R, then what does? Thirdly, can the L597R mutant trans-auto-phosphorylate anyway, or is it completely inactive as a L597R mutant homodimer? Certainly, our *in vitro* assays suggest that this might be the case. We speculate that monomeric L597R captured by Hsp90-CDC37, or by 14-3-3, might be transferred to the RAS-14-3-3 complex, where RAS drives the mutant into a stable and active conformation with CRAF. We suggest that L597R is impaired in kinase activity and, consequently, requires an active partner, such as CRAF (Figure 12).

Data suggests that active RAS is able to induce cRAF-BRAF heterodimerization by exposing 14-3-3 binding sites in the C-terminus of CRAF [53]. That CRAF is required for L597R signaling is also supported by the fact that BRAF mutants, with impaired or intermediate kinase activity, were shown to induce strong activation of CRAF. However, only impaired kinase activity mutants, as seen here with L597R, were shown to be dependent on CRAF for ERK activation [35]. We suggest that because of the increased stability of the L597R mutation towards dimerization, perhaps L597R-CRAF activity cannot now be downregulated, whether by RAS or Hsp90-CDC37. Thus, as an overly stable dimer this could lead to enhanced or sustained signaling and apparent RAS independency, in terms of SOS inhibition and down regulation of overall RAF signaling. In fact, it was recently reported that for some class 2 BRAF mutants, there are variable and overlapping levels of enriched RAS alterations [54]. These authors used variation coexistence between activated RAS and BRAF alterations to support *in vivo* RAS dependency and show that class 2 BRAF alterations have a higher frequency of RAS dependency than class 1 mutants, such as V600E. Our findings show that the double mutant, L597R-V600E and the single mutant, L597R, of BRAF are inactive *in vitro*, which suggests that the L597R mutant may require additional factors for attaining an active signaling state. We suggest this probably involves the RAS-14-3-3 complex together with CRAF. However, whatever the scenario, our findings call for reevaluation of RAS dependency for at least some so-called class 2 BRAF mutations and, as such, the exact mechanism for the activation of L597R still needs to be determined. Thus, we find that mutations in BRAF could potentially influence the dynamics of various BRAF complexes that normally tightly regulate the activity of BRAF. As a result, Hsp90 either preserves V600E kinase domain, by protecting it against proteasomal degradation, or, in the case of L597R, it may promote a dimeric state, probably with CRAF, that is able to signal through MEK in a sustained manner.

The potentially different signaling complexes of BRAF V600E and BRAF L597R could have clinical implications for how to treat tumors driven by each of these driver mutations. Particularly relevant is the fact that CDC37 protects its clients from inhibitor binding, acting as a competitive inhibitor and altering the structure of the kinase, thus blocking kinase inhibitor binding. However, the Hsp90-CDC37 complex-free V600E is susceptible to inhibitors, such as vemurafenib, but it is likely that elevated levels of Hsp90-Cdc37-BRAF V600E, which are largely insensitive to kinase inhibitor, maintain a reservoir of mutant V600E that re-establishes signaling in the normal course of Hsp90 activity. Thus, the combined use of appropriate inhibitors that target the Hsp90-Cdc37-kinase complex, together with BRAF inhibitor, may be advantageous when targeting V600E-driven tumors. In the case of L597R, Cdc37 appears to act as a competitive inhibitor against non-dimerized L597R mutant, and vemurafenib may effectively inhibit L597R signaling as expected [45]. Furthermore, Hsp90 inhibition is effective against murine lung adenocarcinomas driven by the L858R, the equivalent L597R mutation of EGFR [55]. However, recent findings have also identified a sub-group of melanomas, which are driven by BRAF mutants with low-activity and which consequently rely on CRAF signaling [56]. Such mutant cell lines were particularly sensitive to the CRAF specific inhibitor sorafenib. This is an important consideration in targeting BRAF melanoma, and perhaps tumors driven by L597R, as it has been observed that elevated CRAF levels represent a mechanism for acquired resistance to BRAF inhibition [57]. Furthermore, targeting CRAF in a variety of melanoma cell lines was shown to decrease their viability, which appears to be mediated by Bcl-2 inhibition, rather than MAPK inhibition [58]. This may, therefore, provide a clear rationale for not only targeting non-V600E BRAF-driven tumors with BRAF inhibitors, but also targeting the CRAF-dependency of such cell lines. We, therefore, propose that targeting CRAF in L597R-driven tumors combined with BRAF, and perhaps also Hsp90 inhibition, may have a potential therapeutic benefit.

## Figures and Tables

**Figure 1 biomolecules-12-00905-f001:**
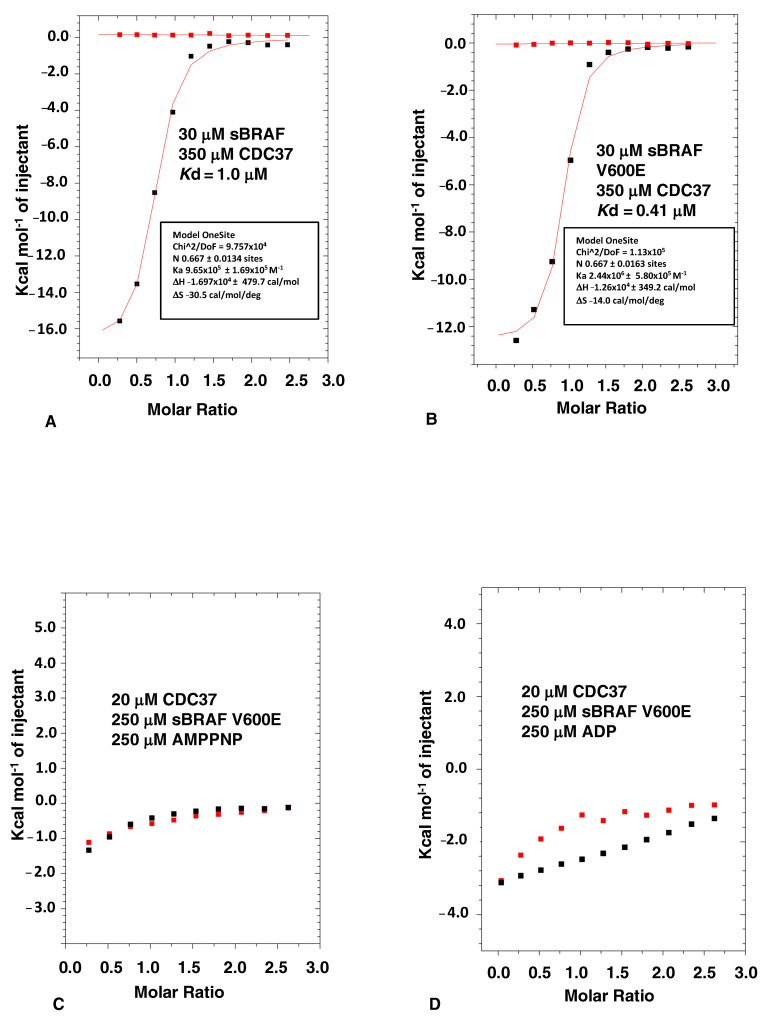
ITC of sBRAF and sBRAF V600E mutant. (**A**) The CDC37 interaction with sBRAF and (**B**) with sBRAF V600E. (**C**) The CDC37 interaction in the presence of AMPPNP with sBRAF V600E and (**D**) ADP with sBRAF V600E. Red markers, represent the heat of dilution and black markers the heat-of-dilution corrected interaction experiment.

**Figure 2 biomolecules-12-00905-f002:**
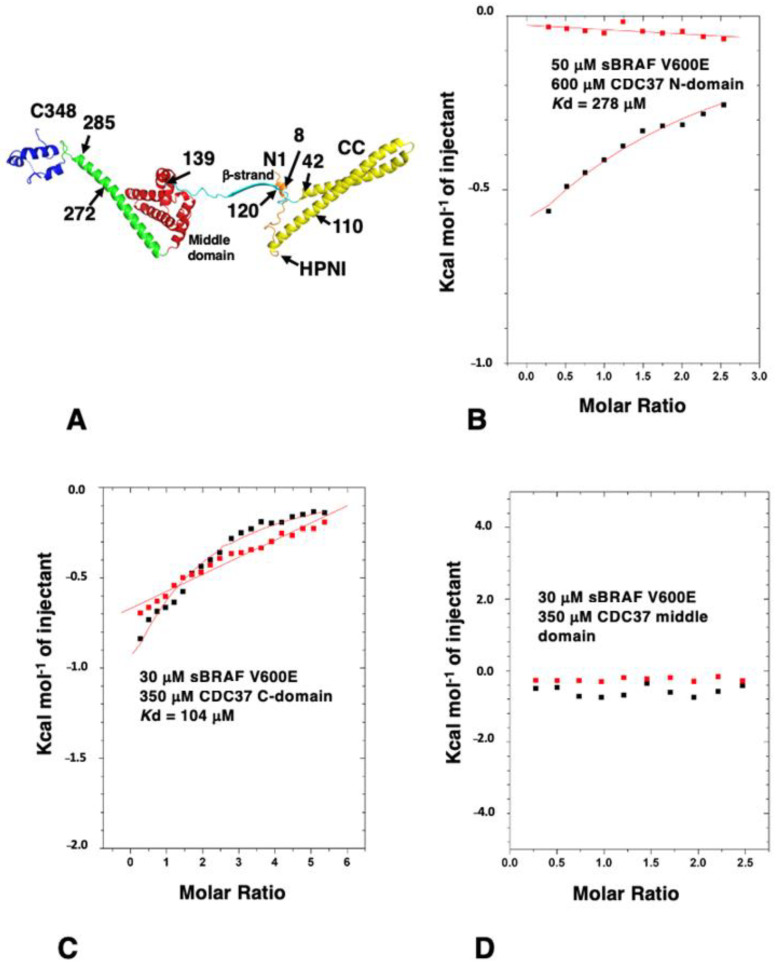
Structure of human CDC37 and ITC using CDC37 domains. (**A**) Individual elements of human CDC37 are colored as follows. orange; residues 1 to 23 including the conserved Ser 13 and HPNI residues; yellow (coiled-coil), residues 24 to 112; cyan, residues 113 to 139 including the beta strand residues 120–129; red, middle domain residues 140–244; green, helix connecting middle- and C-domains residues 245 to 292 and blue, c-terminal domain residues 293 to 348. The structure of residues beyond 348 have not been determined. (**B**) The interaction between sBRAF V600E and the n-terminal domain (residues 1–120), (**C**) with the C-domain (residues 273 to 348) and (**D**) with the middle-domain (residues 148–269) of CDC37. Red markers, represent the heat of dilution and black markers the heat-of-dilution corrected interaction experiment.

**Figure 3 biomolecules-12-00905-f003:**
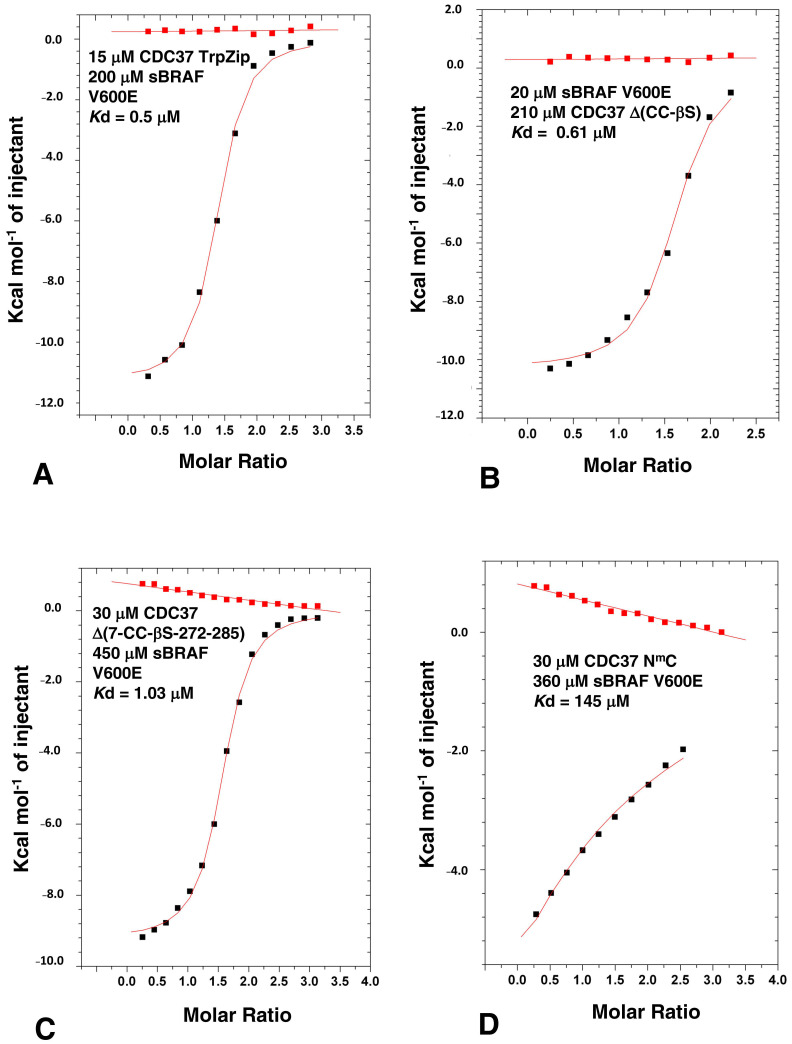
ITC of sBRAF V600E with various CDC37 deletion mutants. (**A**) sBRAF interaction with CDC37 TrpZip, (**B**) with CDC37 Δ(CC-βS), (**C**) with CDC37 Δ(7-CC-βS-272-285) and with (**D**) CDC37 N^m^C. Red markers, represent the heat of dilution and black markers the heat-of-dilution corrected interaction experiment.

**Figure 4 biomolecules-12-00905-f004:**
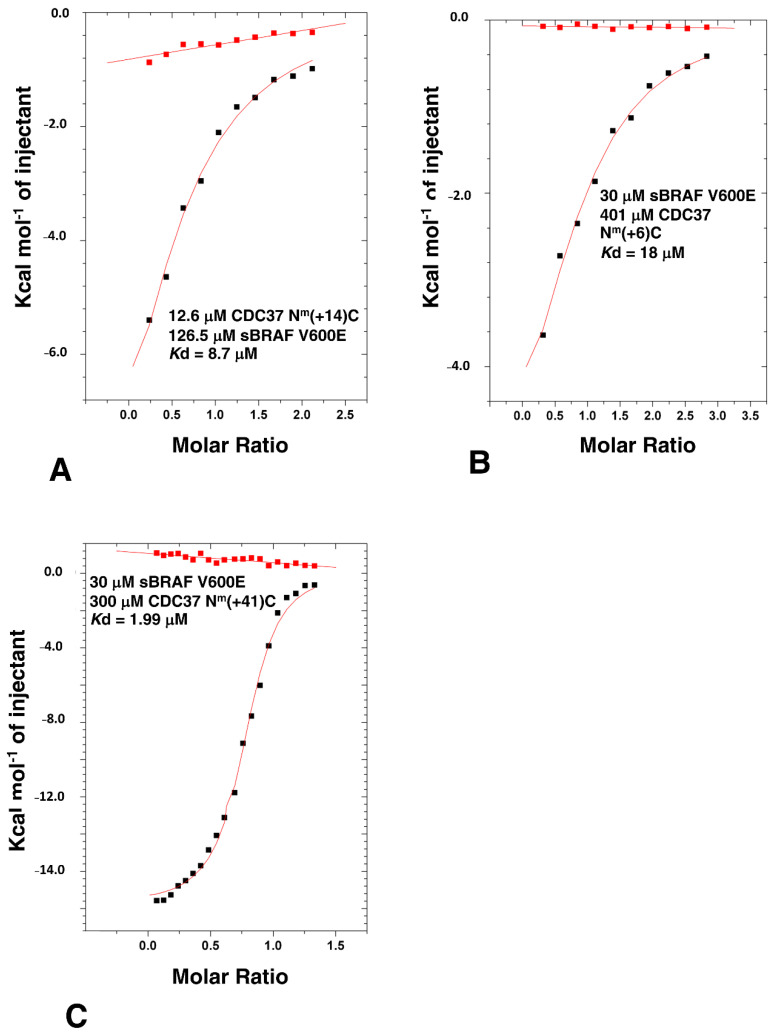
ITC of sBRAF V600E with various CDC37 deletion mutants. (**A**) sBRAF interaction with CDC37 N^m^(+14)C, (**B**) with CDC37 N^m^(+6)C and (**C**) with CDC37 N^m^(+41)C. Red markers, represent the heat of dilution and black markers the heat-of-dilution corrected interaction experiment.

**Figure 5 biomolecules-12-00905-f005:**
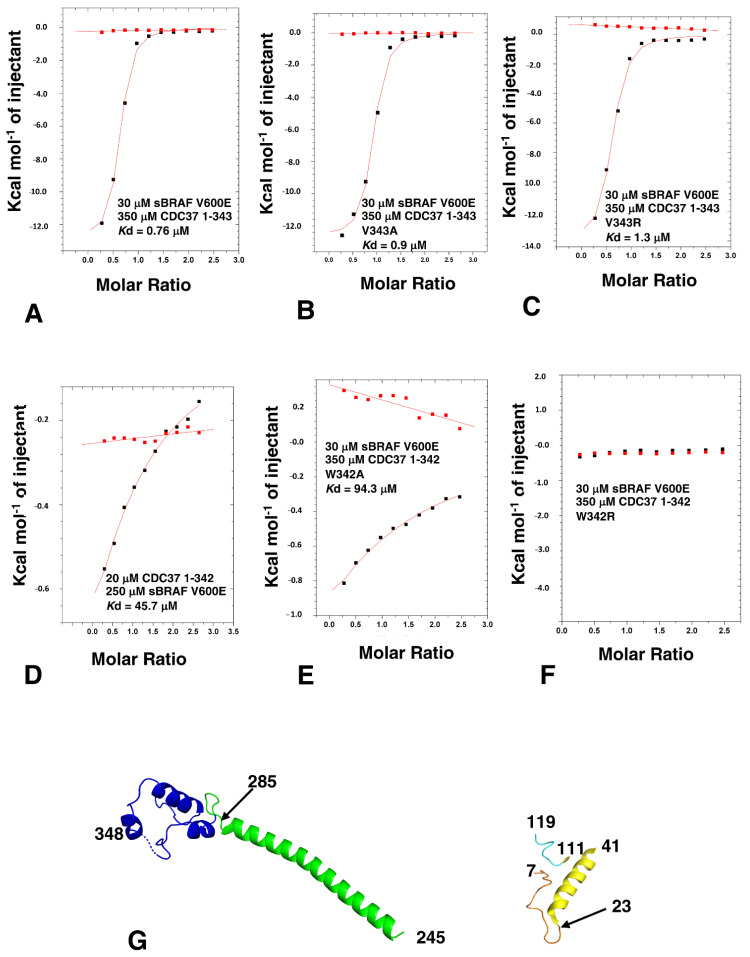
ITC of sBRAF V600E with various CDC37 deletion mutants and the minimal CDC37 elements required for kinase binding. (**A**) sBRAF interaction with CDC37 1–343, (**B**) with CDC37 V343A, (**C**) with CDC37 V343R, (**D**) with CDC37 1-342, (**E**) with CDC37 W342A and (**F**) with CDC37 W342R. Red markers, represent the heat of dilution and black markers the heat-of-dilution corrected interaction experiment. (**G**) Structural elements of CDC37 which can be tethered to approximate towards normal binding with sBRAF. Elements of CDC37 are colored as gold, residues 8–23; yellow, residues 24-41; cyan, 111–119; green, 245–285 and blue, C-terminus residues 286–348.

**Figure 6 biomolecules-12-00905-f006:**
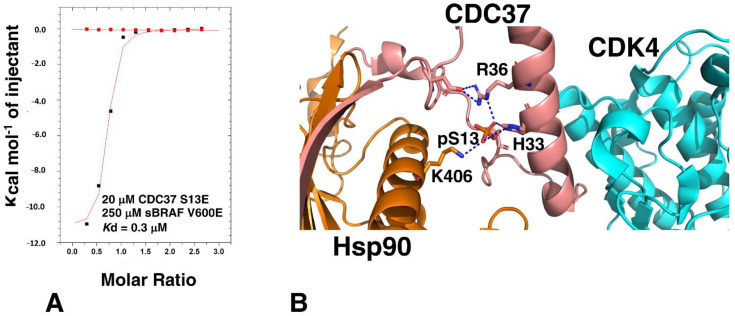
CDC37 pSer 13 is not involved in kinase binding. (**A**) ITC of CDC37-S13E with sBRAF V600E. Red markers, represent the heat of dilution and black markers the heat-of-dilution corrected interaction experiment. (**B**) PyMol cartoon showing the interaction of CDC37 pSER 13 with Hsp90 (gold) and CDC37 (salmon). Direct interactions with Cdk4 (cyan) are not present. Polar interactions are shown by dotted blue lines.

**Figure 7 biomolecules-12-00905-f007:**
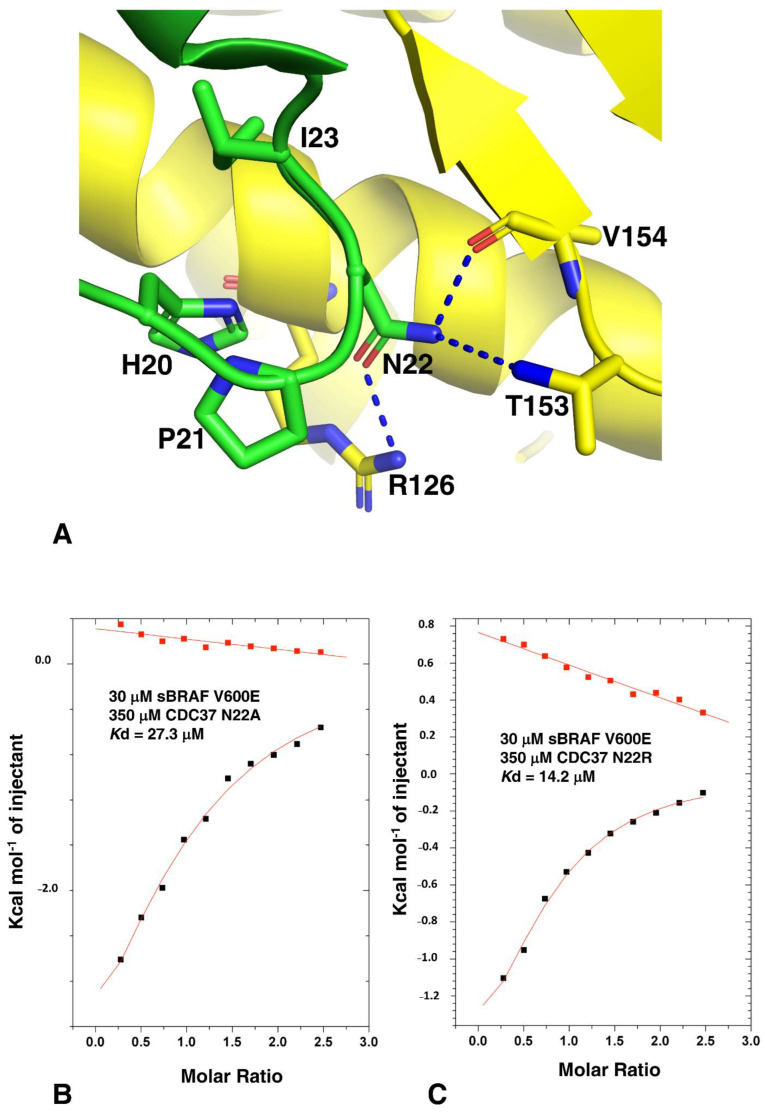
Interaction of CDC37-Asn 22 mutants with Cdk4. (**A**) Possible interaction modelled from the Cryo-EM structure of Hsp90-CDC37-Cdk4 complex. Green, CDC37 and yellow, Cdk4. Polar interactions are shown by dotted blue lines. (**B**) ITC interaction between sBRAF V600E and CDC37 N22A and (**C**) with CDC37 N22R. Red markers, represent the heat of dilution and black markers the heat-of-dilution corrected interaction experiment.

**Figure 8 biomolecules-12-00905-f008:**
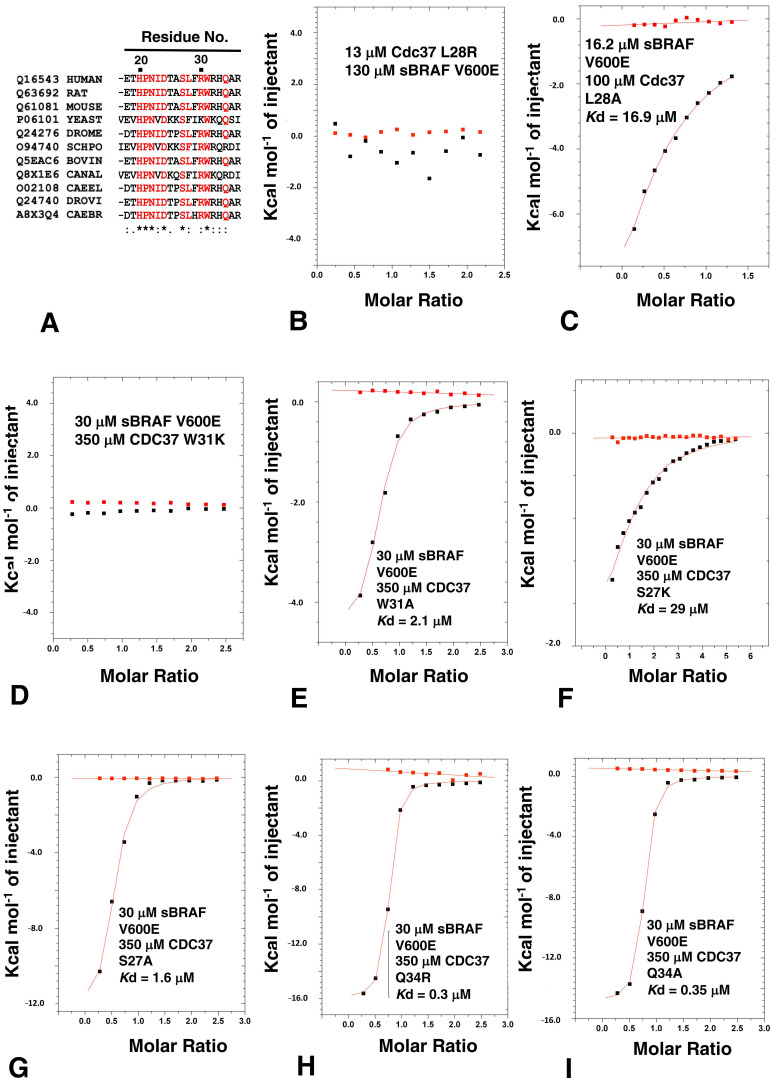
ITC analysis of CDC37 amino acid residues involved in kinase recognition. (**A**) CDC37 alignment with the uniprot accession codes shown together with abbreviation for genus and species. DROME, Drosophila melanogaster (Fruit fly); SCHPO, Schizosaccharomyces pombe (Fission yeast); CANAL, Candida albicans; CAEEL, Caenorhabditis elegans; DROVI, Drosophila virilis (Fruit fly) and CAEBR, Caenorhabditis briggsae. (.), conservation between groups of weakly similar properties; (:), conservation between groups of strongly similar properties and (*), positions that have a single and fully conserved residue. ITC interactions of sBRAF V600E with (**B**) CDC37 L28R, (**C**) with CDC37 L28A, (**D**) with CDC37 W31K and (**E**) with CDC37 W31A. ITC interaction between sBRAF V600 wild type and (**F**) CDC37 S27K, (**G**) with CDC37 S27A, (**H**) with CDC37 Q34R and (**I**) with CDC37 Q34A. Red markers, represent the heat of dilution and black markers the heat-of-dilution corrected interaction experiment.

**Figure 9 biomolecules-12-00905-f009:**
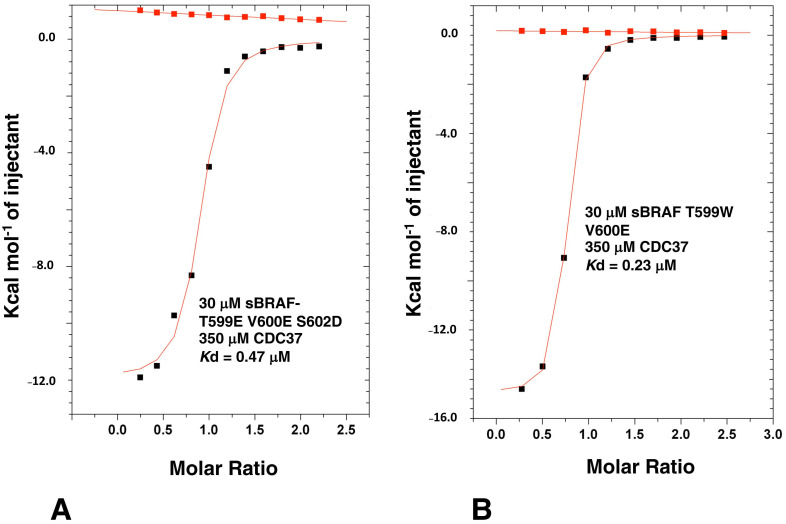
ITC interactions between CDC37 and T-loop mutants of sBRAF. (**A**) Interaction between CDC37 and the triple sBRAF mutant T599E-V600E-S602D and (**B**) with the double mutant T599W-V600E. Red markers, represent the heat of dilution and black markers the corrected interaction experiment.

**Figure 10 biomolecules-12-00905-f010:**
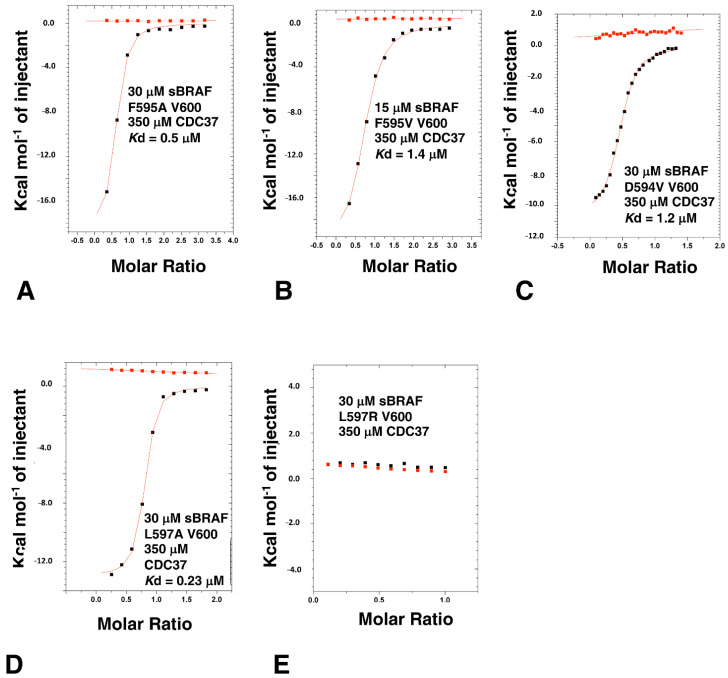
ITC interaction studies with the DFGL mutants of sBRAF. Interactions between CDC37 and (**A**) sBRAF F595A, (**B**) with sBRAF F595V, (**C**) with sBRAF D594V, (**D**) with sBRAF L597A and (**E**) with sBRAF L597R. Red markers, represent the heat of dilution and black markers the heat-of-dilution corrected interaction experiment.

**Figure 11 biomolecules-12-00905-f011:**
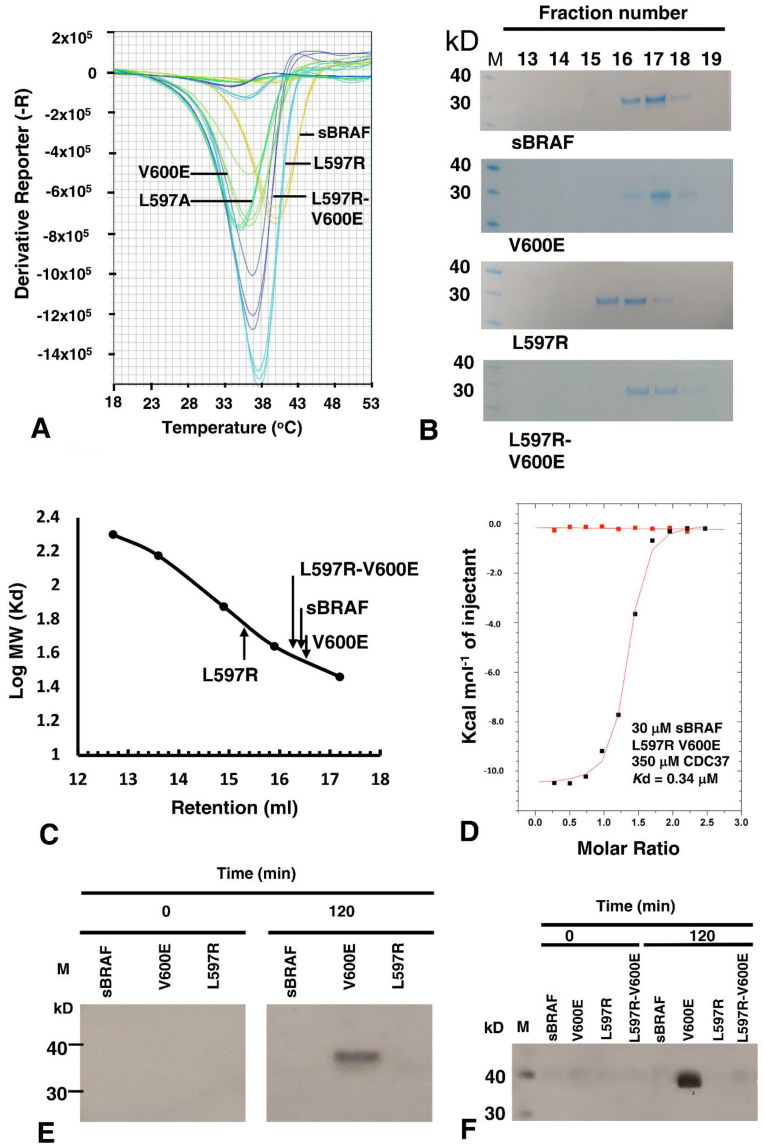
Interaction studies and assays with the L597R mutant of sBRAF. (**A**) Thermal Shift Assay for sBRAF and mutants. In order of decreasing Tm: Orange, sBRAF wild type (Tm = 40 °C); cyan, L587R (Tm = 37.8 °C); blue, L597R-V600E (Tm = 36.8 °C); light green, V600E (Tm = 36.2 °C) and dark green L597A (Tm = 35 °C). (**B**) Gel-filtration fractions from Superdex 200 Increase 10/300 GL column, showing L597R to have a relative molecular mass equivalent to dimeric sBRAF and that V600E disrupts the dimeric nature of the L597R mutant. M, molecular weight markers (kD). (**C**) Estimation of the relative molecular mass of sBRAF mutants, showing L597R to be dimeric in mass. (**D**) ITC between CDC37 and sBRAF L597R-V600E, showing that the V600E mutation restores binding with CDC37 in the L597R background. Red markers, represent the heat of dilution and black markers the heat-of-dilution corrected interaction experiment. (**E**) MEK phosphorylation assays with L597R and V600E, showing that V600E is active and (**F**) with sBRAF L597R showing that is not activated by a V600E mutation. M, molecular weight markers (kD).

**Figure 12 biomolecules-12-00905-f012:**
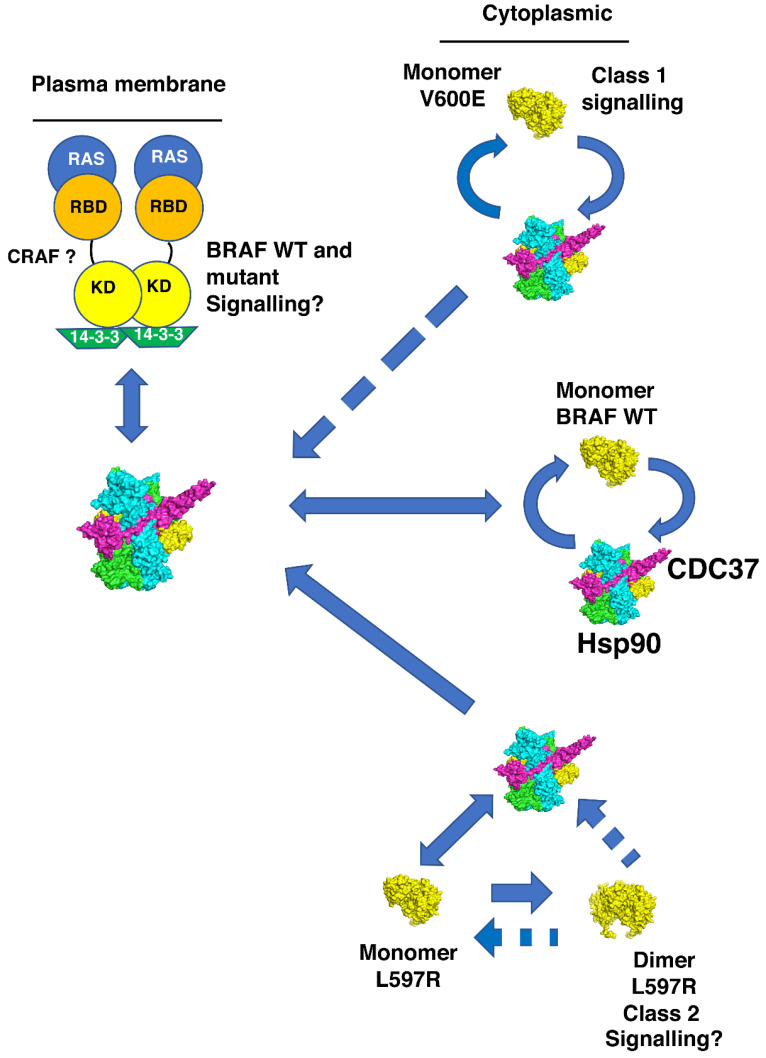
Potential dynamics for the activation of BRAF and the L597R and V600E mutants by Hsp90 and RAS. Wild type (WT) BRAF is transported to the RAS complex by Hsp90-CDC37, where its activity is tightly regulated. V600E is active as a monomer and appears to associate preferentially with Hsp90 complex, which provides cellular stability. Its active conformation may thus limit its entry to the RAS complex, as early stages of this complex contain BRAF in an autoinhibited conformation. Newly made L597R monomers may be assembled into dimers (perhaps with CRAF) following Hsp90-CDC37 association and activation, where they may then remain resistant to disassembly by the chaperone system. BRAF L597R may also be transferred to the RAS complex where its enhanced stability as a dimer (probably with CRAF) alters the down regulatory mechanism of the BRAF L597R-CRAF kinase heterocomplex. (?—in the figure), represents signaling that needs to be further defined for BRAF L597R kinase. (CRAF?), represents the potential for CRAF replacing one of the BRAF monomers of the dimerized state at the plasma membrane. Solid blue arrows show potential flow of BRAF to the Hsp90 and RAS complex, whereas broken arrows show possible restricted movement relative to wild type BRAF. Delivery of BRAF, its mutants and CRAF could also take place with 14-3-3, but this has been omitted from the figure for clarity. KD, BRAF kinase domain and RBD, RAS binding domain of BRAF. Structural models are represented as green and cyan, Hsp90, magenta, CDC37 and yellow BRAF.

## Data Availability

Raw data for ITC and the thermal shift assay can be found at https://doi.org/10.25377/sussex.c.6061241. Constructs used in this study in Appendix A.

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
