# Peer review of "Recognition of BRAF by CDC37 and Re-Evaluation of the Activation Mechanism for the Class 2 BRAF-L597R Mutant"

_biomolecules, 2022, doi:10.3390/biom12070905_

Round 1

Reviewer 1 Report

In this article, the authors examine the impact of oncogeneic BRAF mutations on BRAF dimerization, activity, and ability to bind the co-chaperone Cdc37. They demonstrate through a series of well-crafted in vitro experiments that in contrast to the hyperactive V600E mutation, the BRAF L597R mutation promotes dimerization and an inability to bind Cdc37. This work provides a framework for understanding the connection between molecular chaperones and the activity of hyperactive kinases in cancer. The methods are solid, the manuscript is well-written and appropriate efforts have been taken to ensure high data quality. I highly recommend this exciting work for publication. Minor comments:

1)    Phospho-mimic mutation of kinase activation loops do not always produce an increase in kinase activity (as in the case of ERK2). Can the authors provide a reference for T599E and S602E producing a hyperactive kinase?

2)    The authors cite Taha et al for the idea that L597R is hyperactive, but this is actually a review paper. Looking at the primary literature, I cannot find any evidence that this mutant is hyperactive in regards to its kinase activity. The closest I could find was https://pubmed.ncbi.nlm.nih.gov/26343582/ but this only examines L597Q. Could the authors provide a primary literature source that tests L597R for kinase activity? If there isn’t one, this may actually explain the authors result in Fig. 11F, where activity is not observed for L597R (in contrast to V600E). This result makes sense-if Cdc37 cannot bind L597R, then this mutant would display lowered kinase activity.

3)    The final figure, while a welcome addition is a bit hard to decipher for a non-expert. The basic concept of this paper is that BRAF needs to become monomeric to bind Cdc37 and become active as a kinase. I would suggest simplifying the figure by switching the structures to basic shapes that are labeled Cdc37, BRaf etc. The arrow notation could also be simplified by removing different colors and duplicate arrows.

Author Response

I would like to thank you for your time and very useful comments

1. Phosphomimic mutation of kinase activation loops do not always produce an increase in kinase activity (as in the case of ERK2). Can the authors provide a reference for T599E and S602E producing a hyperactive kinase?

This is a good point, but we say fully active and did not mean to suggest hyperactive. T599E-V600E-S602D is shown to be active in supplementary data figure 4 of one of our own papers. I have modified the text to make its activity clear, which is similar to that of V600E. I have used the following reference:

Polier S, Samant RS, Clarke PA, Workman P, Prodromou C, Pearl LH. ATP-competitive inhibitors block protein kinase recruitment to the Hsp90-Cdc37 system [published correction appears in Nat Chem Biol. 2013 Jun;9(6):406]. Nat Chem Biol. 2013;9(5):307-312. doi:10.1038/nchembio.1212

2. The authors cite Taha et al for the idea that L597R is hyperactive, but this is actually a review paper. Looking at the primary literature, I cannot find any evidence that this mutant is hyperactive in regards to its kinase activity. The closest I could find was https://pubmed.ncbi.nlm.nih.gov/26343582/ but this only examines L597Q. Could the authors provide a primary literature source that tests L597R for kinase activity? If there isn’t one, this may actually explain the authors result in Fig. 11F, where activity is not observed for L597R (in contrast to V600E). This result makes sense-if Cdc37 cannot bind L597R, then this mutant would display lowered kinase activity.

This is an excellent point you have raised. The evidence as it turns out is indirect. It looks at the abundance of pMEK in L597R cells and activation of Erk is the only indirect evidence for L597R activity. It was also shown that pCRAF levels in these cells was low, non-the-less we cannot say if pMEK abundance is due to L597R dimers or L597R-CRAF dimers. Ultimately though, these observations are consistent with our findings in that L597R is inactive, but relies on CRAF dimerization for activity.

We have consequently rewritten this section and updated the paper accordingly.

The references showing this work is as follows and the results are actually in the supplementary section of the paper. We have therefore revised our paper to make this observation clear and thank you for this very important point that you raised.

Dahlman KB, Xia J, Hutchinson K, et al. BRAF(L597) mutations in melanoma are associated with sensitivity to MEK inhibitors. Cancer Discov. 2012;2(9):791-797. doi:10.1158/2159-8290.CD-12-0097

and

BRAF mutants evade ERK dependent feedback by different mechanisms that determine their sensitivity to pharmacologic inhibition

Zhan Yao, Neilawattie M. Torres, Anthony Tao, Yijun Gao, Lusong Luo, Qi Li, Elisa de Stanchina, Omar Abdel-Wahab, David B. Solit, Poulikos Poulikakos and  Neal Rosen. Cancer Cell. 2015 Sep 14; 28(3): 370–383. 

3)    The final figure, while a welcome addition is a bit hard to decipher for a non-expert. The basic concept of this paper is that BRAF needs to become monomeric to bind Cdc37 and become active as a kinase. I would suggest simplifying the figure by switching the structures to basic shapes that are labelled Cdc37, BRaf etc. The arrow notation could also be simplified by removing different colours and duplicate arrows.

Thank you for this suggestion. We have taken this advice and simplified the figure as requested, but maintained structures where components for these are easy to understand, but also added labels to help towards this. We feel this has greatly enhanced the figure and thank you for this suggestion.

Reviewer 2 Report

Title: Recognition of BRAF by CDC37 and Re-evaluation of the Activation mechanism for the Class 2 BRAF-L597R Mutant Authors: Dennis M. Bjorklund, R. Marc L. Morgan, Jasmeen Oberoi, Katie L. I. M. Day, Panagiota A. Galliou , Chrisostomos Prodromou Comments:   This is a well-performed study that has revealed a set of novel interesting data. The submitted manuscript is well written and nicely illustrated. I think that this manuscript may be acceted in the present form. I would only recommend to discuss a bit more the biological meaning/consequences of the elucidated mechanism involving BRAF mutations.        

Author Response

I would like to thank this reviewer for the time and useful suggestions.

  1. I would only recommend to discuss a bit more the biological meaning/consequences of the elucidated mechanism involving BRAF mutations.  

We have added an extended discussion on the clinical implications of the L597R and V600E mutations. Thank you for your suggestion as we feel this has improved the manuscript as a whole and given it more relevant meaning.

Thus:

The potentially different signelling complexes of BRAF V600E and BRAF L597R could have clinical implications of how to treat tumours driven by each of these driver mutations. Particularly relevant is the fact that CDC37 protects its clients form inhibitor binding, acting as a competitive inhibitor and altering the structure of the kinase, thus blocking kinase inhibitor binding. However, the Hsp90-CDC37 complex-free V600E is susceptible to inhibitors such as vemurafenib, but it is likely that elevated levels of Hsp90-Cdc37-BRAF V600E, which are largely insensitive to kinase inhibitor, maintain a reservoir of mutant V600E that re-establishes signalling in the normal course of Hsp90 activity. Thus, the combined use of appropriate inhibitors that target the Hsp90-Cdc37-kinase complex together with BRAF inhibitor maybe advantageous when targeting V600E driven tumours. In the case of L597R, Cdc37 appears to act as a competitive inhibitor against non-dimerised L597R mutant and vermurafenib can effectively inhibit L597R signalling as expected (Dahlman et al., 2012).  Furthermore, Hsp90 inhibition is effective against murine lung adenocarcinomas driven by the L858R, the equivalent L597R mutation of EGFR (Shimamura et al, 2008). However, recent findings have also identified a sub-group of melanomas, which are driven by BRAF mutants with low-activity and consequently rely on CRAF signalling (Smalley et al, 2009). Such mutant cell lines were particularly sensitive to the CRAF specific inhibitor sorafenib. This is an important consideration in targeting BRAF melanoma, and perhaps tumours driven by L597R, as it has been observed that elevated CRAF levels represent a mechanism for acquired resistance to BRAF inhibition (Montagut et al, 2008). Furthermore, targeting CRAF in a variety of melanoma cell lines was shown to decrease their viability, which appears to be mediated by Bcl-2 inhibition rather than MAPK inhibition (Jilaveanu et al, 2009). This may therefore provide a clear rationale for not only targeting non-V600E BRAF driven tumours with BRAF inhibitors, but also targeting the CRAF-dependency of such cell lines. We therefore propose that targeting CRAF in L597R driven tumours combined with BRAF and perhaps also Hsp90 inhibition, may have a potential therapeutic benefit.